# Human Milk Oligosaccharide Profile Variation Throughout Postpartum in Healthy Women in a Brazilian Cohort

**DOI:** 10.3390/nu12030790

**Published:** 2020-03-17

**Authors:** Ana Lorena Ferreira, Ronaldo Alves, Amanda Figueiredo, Nadya Alves-Santos, Nathalia Freitas-Costa, Mônica Batalha, Chloe Yonemitsu, Nadia Manivong, Annalee Furst, Lars Bode, Gilberto Kac

**Affiliations:** 1Nutritional Epidemiology Observatory, Department of Social and Applied Nutrition, Institute of Nutrition Josué de Castro, Rio de Janeiro Federal University, Rio de Janeiro 21941-590, Brazil; analorenaferreiira@gmail.com (A.L.F.); ronaldofsalves@gmail.com (R.A.); amanda.cfg@gmail.com (A.F.); nadyahasantos@gmail.com (N.A.-S.); ncristinafc@gmail.com (N.F.-C.); monicaabatalha@gmail.com (M.B.); 2Department of Pediatrics and Mother-Milk-Infant Center of Research Excellence, University of California, San Diego, La Jolla, CA 92161, USA; chloeyonemitsu@yahoo.com (C.Y.); nmanivong@med.unr.edu (N.M.); aloeffler@ucsd.edu (A.F.); lbode@health.ucsd.edu (L.B.); 3Graduate Program in Nutrition, Institute of Nutrition Josué de Castro, Rio de Janeiro Federal University, Rio de Janeiro 21941-590, Brazil

**Keywords:** human milk oligosaccharides, secretor, lactation, human milk composition, body mass index, HPLC, sialyllactose, fucosyllactose

## Abstract

Human milk oligosaccharide (HMO) composition varies throughout lactation and can be influenced by maternal characteristics. This study describes HMO variation up to three months postpartum and explores the influences of maternal sociodemographic and anthropometric characteristics in a Brazilian prospective cohort. We followed 101 subjects from 28–35 gestational weeks (baseline) and throughout lactation at 2–8 (visit 1), 28–50 (visit 2) and 88–119 days postpartum (visit 3). Milk samples were collected at visits 1, 2 and 3, and 19 HMOs were quantified usinghigh-performance liquid chromatography with fluorescence detection (HPLC-FL). Friedman post-hoc test, Spearman rank correlation for maternal characteristics and HMOs and non-negative matrix factorization (NMF) were used to define the HMO profile. Most women were secretors (89.1%) and presented high proportion of 2′-fucosyllactose (2′FL) at all three sample times, while lacto-N-tetraose (LNT, 2–8 days) and lacto-N-fucopentaose II (LNFPII, 28–50 and 88–119 days) were the most abundant HMOs in non-secretor women. Over the course of lactation, total HMO weight concentrations (g/L) decreased, but total HMO molar concentrations (mmol/L) increased, highlighting differential changes in HMO composition over time. In addition, maternal pre-pregnancy body mass index (BMI) and parity influence the HMO composition in healthy women in this Brazilian cohort.

## 1. Introduction

Human milk is a conditionally complete food, allowing infants to reach adequate growth and development [1,2,3,4]. The paramount benefits for both mother and child health include reduced risk of breast cancer and diabetes for mothers and lower infectious morbidity and mortality for children [5]. The World Health Organization (WHO) has recommended since 2001 the adoption of exclusive breastfeeding for the first six months of life [6,7]. These health benefits have been related to the composition of human milk [8,9].

Human milk is composed of macronutrients (carbohydrates, proteins and fat), micronutrients (vitamins and minerals) and bioactive compounds (cytokines, hormones, growth factors, among others) [8]. Oligosaccharides, part of the carbohydrate fraction, are the 3rd most abundant solid component of human milk (after lactose and fat) [10]. Human milk oligosaccharides (HMOs) are complex unconjugated glycans that are synthesized from lactose, can be by elongated by lacto-N-biose or N-acetyllactosamine disaccharide units, and modified by fucose or sialic acid [10]. More than 150 structurally distinct HMOs have been identified, however, less than 20% of HMOs by number comprise approximately 90% of total HMO composition by concentration [11,12]. HMO concentrations range from 20–25 g/L in colostrum to 5–15 g/L in mature milk [10,13,14].

Maternal genetic features play an important role in HMO composition and variation. Single nucleotide polymorphisms (SNPs) in the secretor gene, encoding for the enzyme fucosyltransferase-2 (FUT2), and the Lewis gene, encoding for fucosyltransferase-3 (FUT3), contribute a large part to the variation in the HMO composition between different women [15,16]. Women with an active FUT2 enzyme are secretors, whose milk contains a high amount of α1-2-fucosylated HMOs like 2′fucosyllactose (2′FL) and lacto-N-fucopentaose I (LNFP I). In contrast, women with specific FUT2 SNPs that introduce a premature stop-codon and abolish the expression of the FUT2 enzyme, are non-secretors and their milk has almost no 2′FL or LNFP1 [10,17,18,19]. In addition to these genetically determined factors, studies have shown that HMO concentrations vary throughout lactation [14,20,21], but also by geographic location, parity, maternal age, weight, body mass index (BMI), mode of delivery and other environmental factors such as seasonality [21,22,23,24,25].

HMOs play important roles in infant health and development outcomes [26,27,28]. Higher HMO diversity is associated with lower total body weight and lower body fat percentage in children [29]. HMOs can act as substrate for specific and potentially health-promoting bacteria and modulate the infant intestinal microbiota composition [30,31,32,33].

Several studies have described the concentrations and variations of HMOs during lactation [20,21,34,35]. However, few investigations have considered a longitudinal design during the first months postpartum and investigated the association between modifiable maternal characteristics and HMO composition. Furthermore, to the best of our knowledge, there are no longitudinal studies during this period in healthy Brazilian or Latin-American women.

This study describes the variation of HMO concentrations at 2–8, 28–50 and 88–119 days postpartum and explores the role of maternal sociodemographic and anthropometric characteristics on HMO composition among healthy women followed in a cohort study in Rio de Janeiro, Brazil.

## 2. Materials and Methods

### 2.1. Study Design and Eligibility Criteria

This study is part of a prospective cohort conducted in a Public Health Care Center in Rio de Janeiro, Brazil. Pregnant women were invited to participate if they met the following eligibility criteria: 18 to 40 years, 28th to 35th weeks of gestation, live in the programmatic area (neighborhood), intended to remain living within the catchment area after the child’s birth, free of infectious and chronic diseases (except obesity) and without twin pregnancy.

The recruitment period lasted from January 2017 to April 2019. A non-probabilistic sample was followed at baseline and five visits took place at 3rd trimester of pregnancy (baseline), 2–8 days (visit 1), 28–50 days (visit 2), 88–119 days (visit 3), 6 months (visit 4) and 12 months postpartum (visit 5). Specifically, this manuscript used data and human milk samples from baseline, and 2–8, 28–50 and 88–119 days. Women that developed gestational diabetes, pre-eclampsia or delivered a stillborn were excluded.

The Research Electronic Data Capture (REDCap), an online platform with robust data entry tool, that allows the management of studies with different types of design, was used to collect information and to create a database [36].

### 2.2. Human Milk Data and Oligosaccharides Analysis

Human milk samples were collected at 2–8 days, 28–50 and 88–119 days according to the Brazilian Network of Human Milk Banks protocol [37]. The milk was collected preferably in the morning and after breakfast. A general guideline on massage techniques and hand expression of breast milk to prevent breast engorgement and mastitis was provided. Afterwards, the participant was counseled for personal protective equipment use (face protection masks and head cover). The investigators performed the same procedures and used nitrile gloves without talcum powder, which reduces the possibility of samples contamination [38]. Investigators were available to assist participants at the time of collection whenever needed.

The mother performed manual milk extractions directly in 50 mL sterile, ribonuclease (RNAse) and deoxyribonuclease (DNAse)-free and non-pyrogenic falcon tubes. At 2–8 days, 5 mL of milk were collected, and were later divided into 5 aliquots of 1 mL. At 28–50 and 88–119 days, 17 mL were collected, which were subdivided into 1 aliquot of 5 mL and 12 aliquots of 1 mL. The samples were processed immediately after collection. Human milk was stored in sterile, RNAse and DNAse-free and non-pyrogenic tubes; refrigerated at −20 °C, transported in a temperature-controlled box (−1 °C to −5 °C), and finally stored in a freezer at −80 °C until analysis.

The samples were shipped to the University of California, San Diego on dry ice in a thermal box to ensure that temperatures were kept close to −80 °C. High-performance liquid chromatography with fluorescence detection (HPLC-FL) was used to characterize HMOs in breast milk, as previously described [39]. Human milk was spiked with raffinose (a non-HMO carbohydrate) as an internal standard to allow for absolute quantification. Oligosaccharides were extracted by high-throughput solid phase extraction over C18 and Carbograph microcolumns and fluorescently labeled with 2-aminobenzamide (2AB). Labeled oligosaccharides were analyzed by HPLC-FL on an amide-80 column (15 cm length, 2 mm inner diameter, 3 μm particle size; Tosoh Bioscience) with a 50 mmol/L ammonium formate–acetonitrile buffer system. Separation was performed at 25 °C and monitored with a fluorescence detector at 360 nm excitation and 425 nm emission. Peak annotation was based on standard retention times and mass spectrometric analysis on a Thermo LCQ Duo Ion trap mass spectrometer, equipped with a Nano-ESI-source. The absolute quantification of the following 19 HMOs was determined based on individual standard response curves and in relation to the internal raffinose standard: 2′FL, 3-fucosyllactose (3FL), 3′-sialyllactose (3′SL), 6′-sialyllactose (6′SL), difucosyllactose (DFLac), difucosyllacto-N-hexaose (DFLNH), difucosyllacto-N-tetrose (DFLNT), disialyllacto-N-hexaose (DSLNH), disialyllacto-N-tetraose (DSLNT), fucodisialyllacto-N-hexaose (FDSLNH), fucosyllacto-N-hexaose (FLNH), lacto-N-fucopentaose (LNFP) I, LNFP II, LNFP III, lacto-N-hexaose (LNH), lacto-N-neotetraose (LNnT), lacto-N-tetrose (LNT), sialyl-lacto-N-tetraose b (LSTb), and sialyl-lacto-N-tetraose c (LSTc). Secretor status was determined by presence or near-absence of 2′FL and LNFP I [22,23].

HMO-bound sialic acid and HMO-bound fucose were calculated as the sum of all sialic acid and fucose moieties bound to the measured HMOs, respectively. Small HMOs were defined by grouping 2′FL, 3FL, 3′SL and 6′SL concentrations. We considered the HMOs extended structures type I (Gal (β1-3) GlcNAc, LactoN-biose) and type II (Gal (β1-4) GlcNAc, N-Acetyllactosamine), to classify these groups: type 1 was calculated with sum of LNT, LNFP I, LNFP II, LSTb and DSLNT and type 2 with LNnT, LNFP III and LSTc. The alpha linkages groups were classified as fucosylated, with Alpha 1,2 (LNFP I and 2′FL) or Alpha 1,3 (LNFP III and 3-fucosyllactose), or as sialylated Alpha 2,6 (LSTc and 6′SL) [40,41].

### 2.3. Co-Variables

Maternal height was measured with a stadiometer (Altura Exata, Belo Horizonte, Brazil), and weight, with weighing scales (Seca 704, Hamburg, Germany). Pre-pregnancy BMI (weight (kg)/height (m)^2^) was calculated using self-reported pre-pregnancy weight and measured height (at 28–50 days) and was categorized as underweight (<18.5 kg/m^2^), normal weight (18.5–24.9 kg/m^2^), overweight (25.0–29.9 kg/m^2^) and obesity (≥30.0 kg/m^2^). Gestational age at birth was based on the first ultrasound, performed prior to 22 (*n* = 98) or 24 (*n* = 1) gestational weeks, or the date of the last menstrual period (*n* = 2) [42].

Gestational weight gain (GWG) was calculated by means of the difference between weight at the last prenatal visit and pre-pregnancy weight (self-reported or retrieved from the prenatal care booklet). The supplements used during pregnancy, and sociodemographic information (maternal age, education and parity) were self-reported and obtained using structured questionnaires. The breastfeeding status was classified according to the WHO definitions [43].

The study was approved by the Research Ethics Committees of the Municipal Secretariat of Health and Civil Defense of the State of Rio de Janeiro (Protocol number: 49218115.0.0000.5275) and of Maternity School of Rio de Janeiro Federal University (Protocol number: 49218115.0.0000.5275), and was carried out following the rules of the Declaration of Helsinki of 1975. Written consent from all participants was obtained freely and spontaneously, after all necessary clarifications were provided.

### 2.4. Statistical Analysis

Statistical analyses were performed with R software, version 3.6 [44]. The variables distribution was evaluated by histograms, skewness and kurtosis measures and a Shapiro–Wilk test (stats package). Outlier values of HMOs concentration were not excluded. Median and interquartile ranges were used to describe maternal sociodemographic and anthropometric characteristics, HMO concentrations and alpha diversity. A Mann–Whitney test was performed in order to evaluate differences between women with and without milk samples. For HMO’s components and variations in HMO group throughout lactation, Friedman’s test and post-hoc analysis for multiple comparisons were performed (using a pairwise multiple comparison of mean ranks—PMCMR package), and a significance level <0.05.

For exploratory analysis, a Spearman rank correlation with heat maps (Corrplot Package) was performed to investigate the correlation between maternal characteristics (age, education, parity, gestational age at birth, GWG, pregnancy iron and folic acid supplement and pre-pregnancy weight, pre-pregnancy BMI) and HMO individuals’ concentrations and alpha diversity index at the different time points. For these analyses, a significance level of <0.01 was adopted and correlations levels were interpreted as: 0.00 to 0.19—very weak correlation, 0.20 to 0.39—weak correlation, 0.40 to 0.69—moderate correlation, 0.70 to 0.89—strong correlation and 0.90 to 1.00—very strong correlation [44].

A stacked bar chart plot (ggplot2 package) was derived to present an absolute concentration and relative abundance of HMOs. The ɑ-diversity index was calculated using Shannon diversity and evenness, inverse Simpson and Pielou evenness (biodiversity R package and vegan package). For index of diversity and evenness comparison in follow-up visits, a Friedman post-hoc test for multiple comparisons was performed.

Non-metric multidimensional scaling (NMDS) was used to define the number of dimensions, preserving the differences to explore patterns of HMO profiles, using the Bray–Curtis dissimilarity matrix (metaMDS in the vegan and ggplot2 package). Subsequently, an analysis of the main components of HMO profiles was performed using the Brunet method, which identified three basic components retained based on the classification estimate. The non-negative matrix factorization (NMF) was used to define potential patterns in the HMO profile data according to postpartum period, BMI nutritional status, parity, and maternal age. The HMO with the highest value on NMF represents the highest probability of that compound contributing to the HMO profile.

Sensitivity analyses were performed using only a subset of 15 women with samples in all time periods.

## 3. Results

### 3.1. Participants and HMOs Concentrations 

A total of 322 women were screened and 147 were recruited. However, 46 women did not collect breast milk samples for the following reasons: 31 missed the visit, 5 failed to collect the samples, 4 women did not breastfeed, 2 collected samples after the predefined time frame of the visit, 3 were excluded (gestational diabetes, pre-eclampsia and delivered a stillborn) and 1 refused to collect. Thus, 101 participants collected milk samples: 52 at 2–8 days (visit 1), 75 at 28–50 days (visit 2) and 46 at 88–119 days (visit 3). Overall, 15 participants had samples for all time points, 42 had samples at two visits and 44 for one visit (Figure 1).

Overall, 89.1% (*n* = 89) of the women were identified as secretors and 10.9% (*n* = 11) as non-secretors. At 2–8 days, 90.4% of mothers were exclusively breastfeeding (*n* = 47) and 9.6 were using infant feeding formulas (*n* = 5). At 28–50 days 67.6% of mothers were exclusively breastfeeding (*n* = 49), 20.3% used infant feeding formulas (*n* = 15), 9.5% were predominantly breastfeeding (*n* = 7) and 2.7% (*n* = 2) were in complementary breastfeeding. At 88–119, 55.8% of mothers were exclusively breastfeeding (*n* = 24), 30.2% used infant formula feeding (*n* = 13) and 14.0% were predominantly breastfeeding (*n* = 6). No differences were observed when comparing women with and without milk samples (Table 1).

The median of total HMO concentrations for all women was 16.66 mmol/L (12.5 g/L) for 2–8 days, 15.48 mmol/L (11.5 g/L) for 28–50 days and 16.79 mmol/L (11.3 g/L) for 88–119 days. Total HMO concentration at 88–119 days was significantly higher than at 2–8 days (*p* ≤ 0.05) and 28–50 days (*p* ≤ 0.05) and was influenced by 3FL and LNFP II (Table 2). The total HMOs concentration according to secretor status through lactation could be found at Appendix A.

Additionally, concerning HMOs grouping, the milk produced at 88–119 days presented a lower median concentration of HMO-bound sialic acid (2.41 mmol/L), type 1 (4.60 mmol/L), type 2 (0.50 mmol/L) and Alpha 2,6 (0.70 mmol/L) compared with 2–8 days and 28–50 days (*p* ≤ 0.05). In contrast, the median of HMO-bound fucose (15.04 mmol/L), small HMO group (8.58 mmol/L) and Alpha 1,2 (1.90 mmol/L) were the highest at 88–119 days compared with 2–8 days and 28–50 days (*p* ≤ 0.05), showing that in later stages, a higher number of smaller molecules was observed (Table 3).

Notably, 2′FL was the most abundant HMO in the milk of secretor women with 32.2% abundance at 2–8 days, 33.8% at 28–50 days, and 25.2% at 88–119 days. In comparison and per definition, 2′FL was nearly absent in the milk of non-secretor women. Instead, LNT was the most abundant HMO at 2–8 days with (30.1%) and LNFPII was the most abundant HMO at 28–50 days (29.4%) and 88–119 days (38.4%) (Figure 2).

Significant differences in HMO concentrations between secretor and non-secretor women (*p* ≤ 0.05) were observed for almost all HMOs except 6′-SL, DSLNH, DSLNT, FLNH, LNFPIII, LNH and LNnT at one or more visits. Among secretor women, the HMOs 3FL (2.24 mmol/L), 3′SL (0.56 mmol/L), DFLac (0.45 mmol/L) and LNFP II (1.55 mmol/L) presented higher values at 88–119 days in comparison with other visits (Figure 3).

### 3.2. Relations between Maternal Anthropometric, Demographic and Reproductive Characteristics and HMO Compositions Concentrations

Spearman correlations varied between low and moderate values. Parity was directly correlated with LNFP II (0.4), DFLNT (0.4), LNH (0.4) and FDSLNH (0.4) at 2–8 days. Pre-pregnancy weight was inversely correlated with 3FL (−0.5), LNFP III (−0.4) and DFLNH (−0.4) and directly correlated with LNnT (0.4) at 2–8 days; inversely correlated with LNFP III (−0.4) and DFLNH (−0.4) at 28–50 days and inversely correlated with 3FL (−0.5) at 88–119 days. Pre-pregnancy BMI was inversely correlated with LNFP III (−0.4) and DFLNH (−0.4) and directly correlated with LNnT (0.4) at 2–8 days; inversely correlated with LNFP III (−0.4) and DFLNH (−0.4) at 28–50 days and inversely correlated with DFLNH (−0.4) and directly correlated with LNnT (0.4) at 88–119 days (Figure 4, Figure 5 and Figure 6).

### 3.3. HMO Diversity and Evenness

The milk produced at 28–50 days presented the highest diversity (Shannon entropy index) and evenness (Shannon and Pielou evenness indexes), with higher median values of indexes compared to the other visits. Additionally, the milk produced in the last visit presented the lowest median for the same indexes of diversity and evenness (Table 4).

### 3.4. HMO Profiles (NMF)

The components that most contributed to HMO profiles at 2–8 days, 28–50 days, and 88–119 days were LNFP II, 3FL and 2′FL, respectively. For all women, the major contribution derived from LNnT (Table 5) and also for age, except the third quartile (Table 6).

3′SL was the HMO that most contributed to the profile of women with pre-pregnancy underweight, while LSTc contributed more for those with normal weight and 2′FL represented the highest contribution for overweight and obese women (Table 7). Overall, 2′FL contributed the most to the HMO profile in women with ≤1 child while LNnT contributed the most to the HMO profile in women with two or more children, according to baseline information (Table 8).

### 3.5. Sensitivity Analyses with Women with Samples in All Time Periods (n = 15)

The majority of the results derived from the sensitivity analyses confirm the findings observed on the complete dataset, such as the smaller total HMO concentration at last visit in g/L, obese women with 2′fucosylactose as the main contributor to the HMO profile and major HMO diversity and evenness at visit 2 (data not shown). However, some minor differences were found. The most abundant HMO on non-secretor women in this subset on the last visit was different from the complete dataset (3-syalillactose). Spearman rank significant correlations were observed between maternal age and FLNH at visit 2 and 3. Finally, maternal characteristics, such as primiparity and normal weight and excess weight, showed a different main contributor HMO in the subgroup, when compared with the complete dataset results.

## 4. Discussion

The present study has several interesting findings. First, differences of HMO concentrations over time were observed. Similar to previous studies [21,29], HMO concentrations in g/L decrease over time postpartum. However, when measured in molar concentrations, HMOs increase over time. This is due to a relative increase in low-molecular HMOs and a relative decrease in high-molecular HMOs over time. The data indicate that looking at weight concentrations in g/L alone provides only a limited picture of HMO composition changes over time. Second, we found that 89.1% of the women were secretors and that 2′FL was the predominant HMO in their milk at all three collection periods. However, for non-secretor women, LNT (at 2–8 days) and LNFPII (at 28–50 days and 88–119 days) were the most abundant HMOs. Third, differences between secretor and non-secretor women were found for 2′FL, 3FL, 3′SL, DFLac, DFLNH, DFLNT, FDSLNH, LNFP I, LNFP II, LNT, LSTb and LSTc—once again highlighting that single nucleotide polymorphisms in a single fucosyltransferase gene affect the synthesis of almost all HMOs, not only those directly fucosylated by the impaired enzyme.

The proportion of 89.1% of secretor women aligns with another recent study conducted in Brazil that identified 87% of women to be secretors [45]. There are significantly more secretors in Latin American countries, such as Brazil and Peru (98%) [22]. In contrast, countries such as Canada (73%), rural Ethiopia (65%), rural Gambia (65%) and Ghana (68%) presented a smaller prevalence of secretor women [22,23].

McGuire et al. [22] studied the same 19 HMOs using our same analytical platform and found total concentrations of 10.1 g/L for Peruvian women at 15–150 days postpartum compared to 12.5 g/L for 2–8 days, 11.5 g/L for 28–50 days, and 11.3 g/L for 88–119 days in the present study. The only other Brazilian study detected 5.57 g/L of HMO at 17–76 days postpartum [46], which is quite different to the present results. Different analytical platforms (quantitative HPLC-FL vs. semi-quantitative LC-MS) and identification and quantification of different HMOs may contribute to the observed differences in HMO data between the two Brazilian cohorts. Results from a systematic review concluded that there is not a standard method to quantify HMOs yet, but also highlighted that mass spectrometry does not provide absolute quantitative measures, which is why we opted to use HPLC-FL with non-HMO internal standard combined with established HMO standard response curves to allow for absolute quantification [47].

We observed that concentrations of DSLNT, LSTc and LNFP I in secretor women decreased over time, which aligns with previously described data in longitudinal studies [48] and in cross-sectional studies with milk samples at different stages of breastfeeding [20,24].

The NMF procedure indicated that different HMOs were responsible for the major contribution of the HMO profiles at different post-partum time points. At 2–8 days, LNFP II presented the major contribution, while, at 28–50 and 88–119 days, 2′FL and 3FL were the major contributors, respectively. Our findings partially agree with McGuire et al. [22] results, that found 2′FL as the main contributor of HMO composition at 20–46 and 79–41 days postpartum.

In this study, pre-pregnancy BMI and pre-pregnancy weight were directly correlated with LNnT at 2–8 days. Unlike our study, McGuire et al. [22] observed that maternal weight and BMI were positively correlated with 2′FL and FLNH, and Samuel et al. [21] described that overweight women presented a lower concentration of LNnT, when compared with normal weight women. Our data revealed that pre-pregnancy BMI and pre-pregnancy weight are inversely correlated with LNFP III at 2–8 days. In contrast, McGuire et al. [22] found maternal weight to be positively correlated with LNFP III. While these studies used the same HMO analytical platform, these differences could indicate a variation between the studied populations and by geographic location. Overall, these discrepancies between cohorts warrant further investigations on how maternal pre-pregnancy BMI, pregnancy weight gain, as well as overall maternal nutrition affect HMO composition and how these changes in HMO composition potentially impact infant health and development.

Azad et al. [23] observed higher concentrations of LNT and LNnT and lower concentrations of 3FL in multiparous women. Likewise, Tonon et al. [45] found a positive correlation between parity and LNT and LNnT and a negative correlation with 3FL. In contrast, Samuel et al. [21] observed higher concentrations of LNnT at 17 days postpartum in primiparous women. We observed that mothers with two children had 2′FL as the main component that contributed to the HMO profile. It is possible that interpregnancy interval may play a role on this association, because this interval was previously described as being associated with changes in breastfeeding practices and could also be linked with human milk composition [49]. However, the mechanism that explains the relationship between parity and HMOs is still unclear and new investigations are necessary.

The sensitivity analyses results were mostly in line with the findings observed for the complete dataset and reaffirm the main conclusions of the study. However, it is important to highlight that some minor differences were observed. It is very likely that these minor differences can probably be attributable to the small sample size of the subset analyzed. 

There are some limitations that pertain to this study. The first is that the small sample size for non-secretors throughout postpartum limits statistical analysis and the comparison of our results with other studies. Additionally, stratifications according to the Lewis blood group were not possible due to this small sample size. Further, during milk collection the breast was not emptied completely. However, some studies have shown small or no variations of HMOs during the feeding. Thus, samples collected in the middle of feeding can be representative of the whole concentration [50,51]. Another limitation was the loss of follow-up during all study visits. Despite these limitations, the present study has important strengths. First, the use of a rigorous analytical platform to quantify HMO concentrations instead of merely measuring relative abundancies. Second, the use of robust statistical method to identify the HMO profile of Brazilian women, with NMF performed to identify patterns that together explain the data set. Third, the longitudinal design used allowed the assessment of prospective changes of HMO concentrations throughout the postpartum period. Last, to the best of our knowledge, this is the first study that has investigated HMOs concentrations at different time points during lactation in Latin American women.

## 5. Conclusions

The results found in the present study suggest that HMO composition varied throughout lactation in Brazilian women, with lower concentrations in g/L at 88–119 days, but an increase in concentrations of low-molecular HMOs at the same period. In addition, maternal pre-pregnancy BMI and parity were associated with HMO composition in healthy women in this Brazilian cohort. These study findings represent an important step toward knowing the profile and variation on HMOs composition in Brazilian women and the characteristics associated with this profile, mainly considering the geographic variation of the HMO concentration in different countries of the world. Additionally, this will be important for future studies, to evaluate the impact of HMO in infants’ outcomes.

## Figures and Tables

**Figure 1 nutrients-12-00790-f001:**
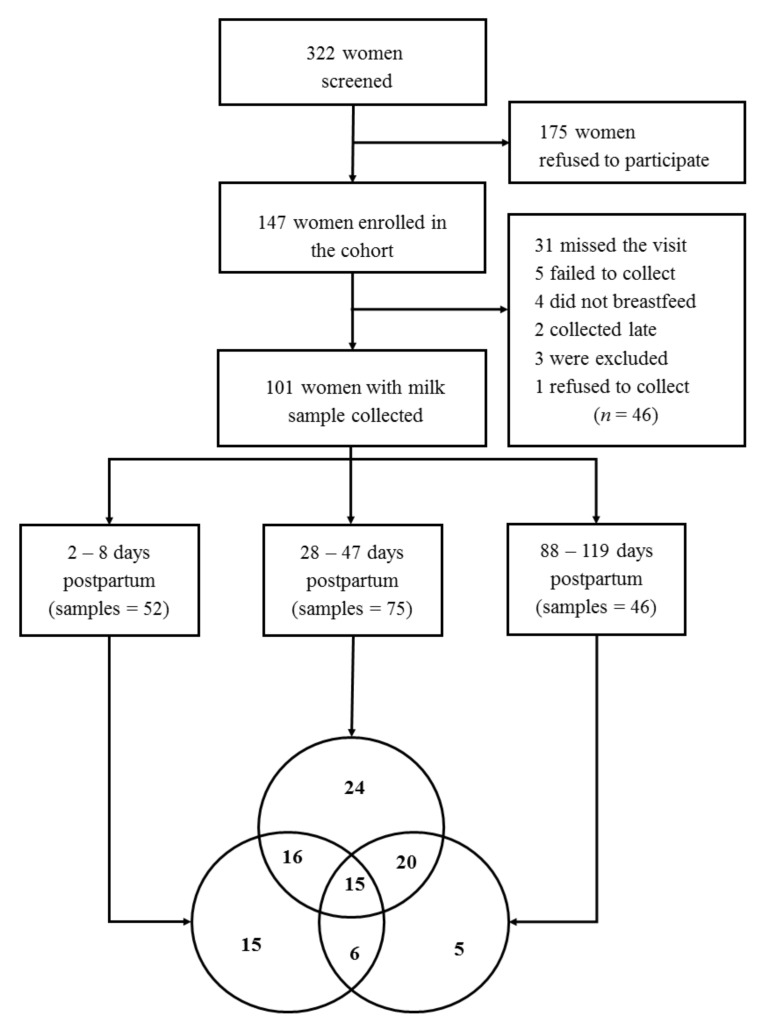
Flow of women and human milk samples of the Brazilian cohort, totaling 101 women with 174 milk samples. The flow intersection represents the women with samples in two or more follow-up visits.

**Figure 2 nutrients-12-00790-f002:**
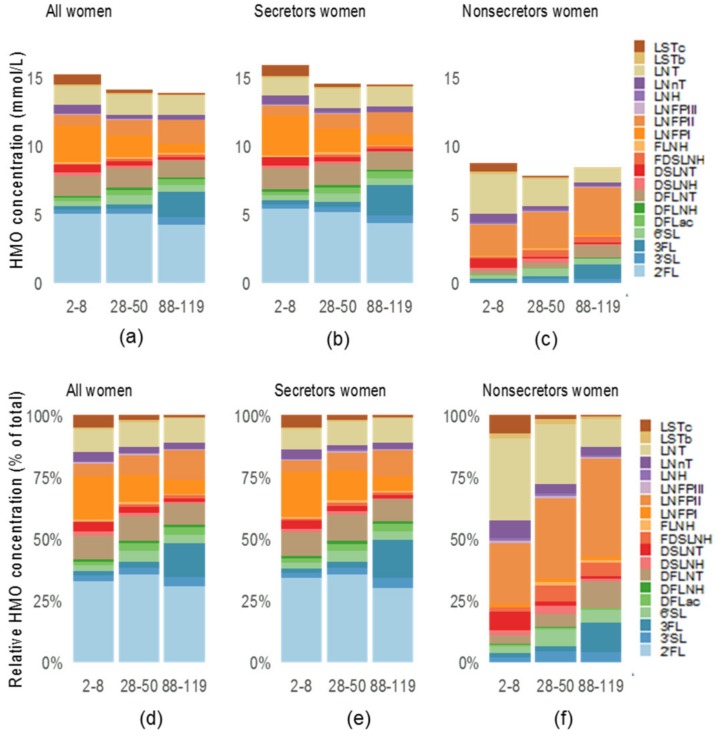
HMO absolute and relative abundance of women from the Brazilian cohort. Rio de Janeiro, Brazil. Median of absolute total HMO (mmol/L) (**a**–**c**) at 2–8, 28–50 and 88–119 days and relative abundance (**d**–**f**) at 2–8, 28–50 and 88–119 days concentrations of all women, according secretor status and follow-up points. DFLac, difucosyllactose; DFLNH, difucosyllacto-N-hexaose; DFLNT, difucosyllacto-N-tetrose; DSLNH, disialyllacto-N-hexaose; DSLNT, disialyllacto-Ntetraose; FDSLNH, fucodisialyllacto-N-hexaose; FLNH, fucosyllacto-N-hexaose; HMO, human milk oligosaccharide; LNFP, lacto-N-fucopentaose; LNH, lacto-N-hexaose; LNnT, lacto-N-neotetraose; LNT, lacto-N-tetrose; LSTb, sialyl-lacto-N-tetraose b; LSTc, sialyl-lacto-N-tetraose c; 2′FL, 2′-fucosyllactose; 3FL, 3-fucosyllactose; 3′SL, 3′-sialyllactose; 6′SL, 6′-sialyllactose.

**Figure 3 nutrients-12-00790-f003:**
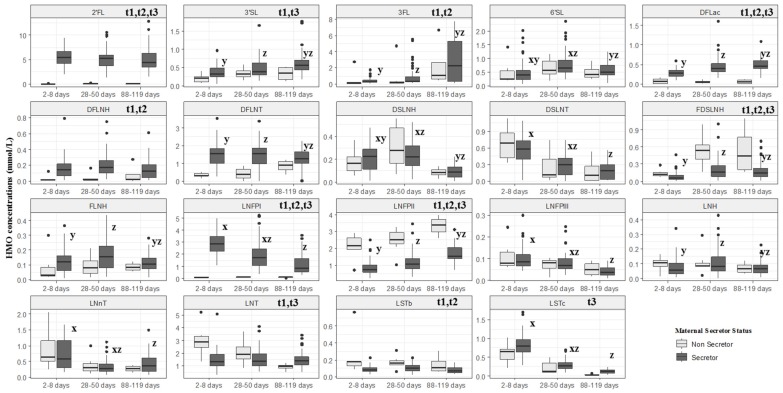
Box plot of secretor and non-secretor HMOs concentration of women from Brazilian cohort. Rio de Janeiro, Brazil. The description t1, t2 and t3 indicate differences between secretor and non-secretor concentration of according to Mann–Whitney, *p* < 0.05 for each time. ^x^ indicates that the HMO concentration at visit 1 is significantly different (*p* < 0.05) from the concentration at visit 2, ^y^ indicates that the HMO concentration at visit 1 is significantly different (*p* < 0.05) from the concentration at visit 3 and ^z^ indicates that the HMO concentration at visit 2 is significantly different (*p* < 0.05) from the concentration at visit 3 according to Friedman post-hoc test. A comparison between the times of non-secretor women was not performed.

**Figure 4 nutrients-12-00790-f004:**
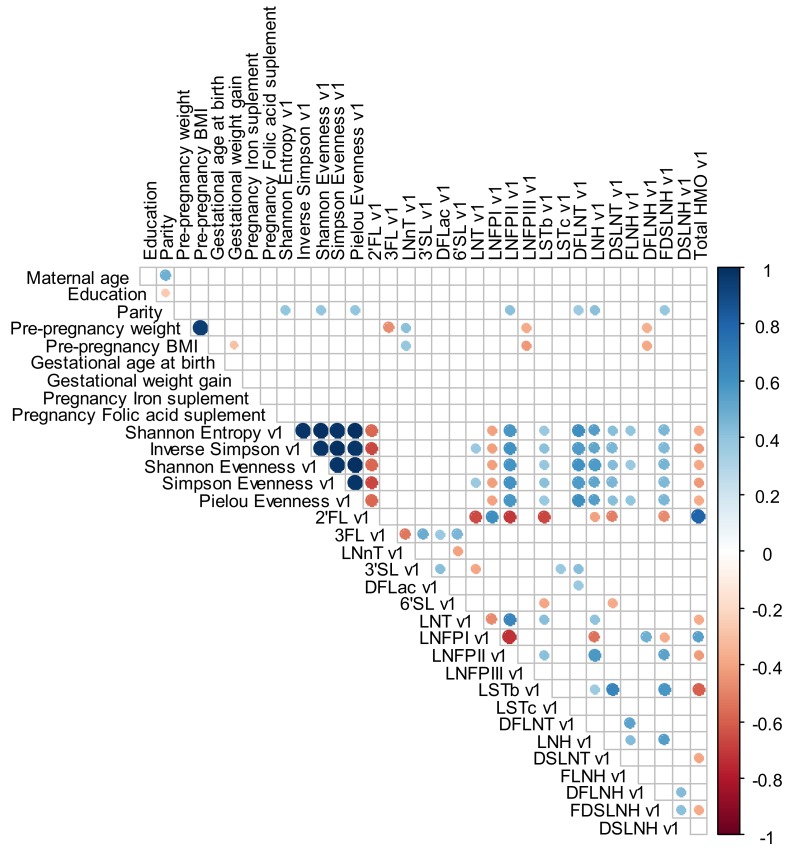
Spearman rank correlations between maternal demographic, anthropometric, reproductive and supplementary variables and individual HMO concentration in mmol/L and diversity and evenness at 2–8 days. The circle only becomes visible for the correlations that present a level of significance ≤0.01. The blue color indicates a positive and the red color a negative correlation. BMI: body mass index; DFLac, difucosyllactose; DFLNH, difucosyllacto-N-hexaose; DFLNT, difucosyllacto-N-tetrose; DSLNH, disialyllacto-N-hexaose; DSLNT, disialyllacto-N-tetraose; FDSLNH, fucodisialyllacto-N-hexaose; FLNH, fucosyllacto-N-hexaose; HMO, human milk oligosaccharide; LNFP, lacto-N-fucopentaose; LNH, lacto-N-hexaose; LNnT, lacto-N-neotetraose; LNT, lacto-N-tetrose; LSTb, sialyl-lacto-N-tetraose b; LSTc, sialyl-lacto-N-tetraose c; 2′FL, 2′-fucosyllactose; 3FL, 3-fucosyllactose; 3′SL, 3′-sialyllactose; 6′SL, 6′-sialyllactose.

**Figure 5 nutrients-12-00790-f005:**
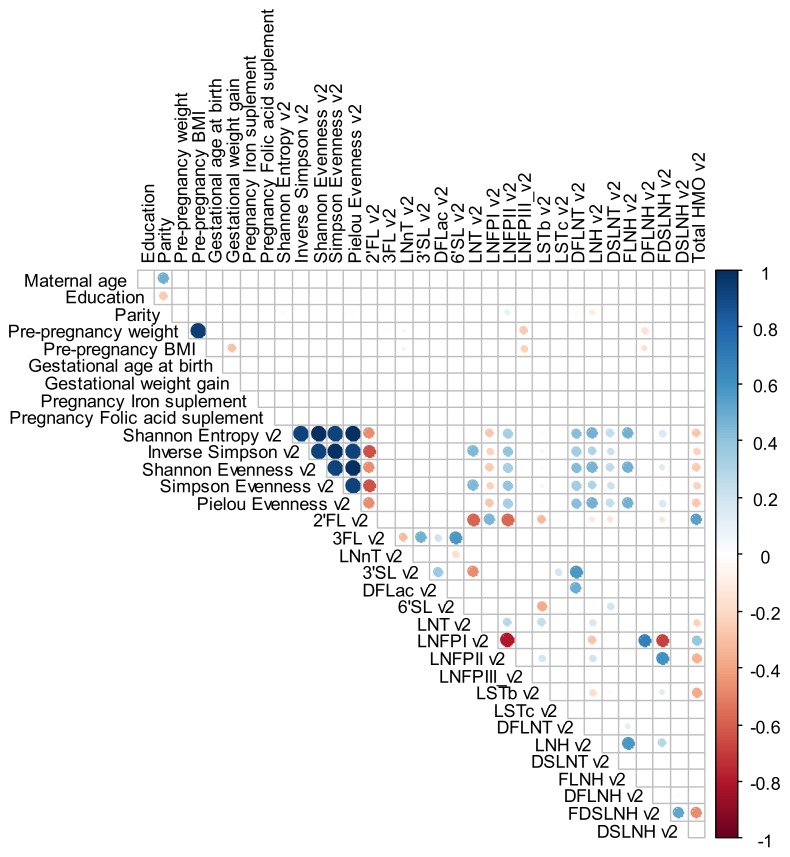
Spearman rank correlations between maternal demographic, anthropometric, reproductive and supplementary variables and individual HMO concentration in mmol/L and diversity and evenness at 28–50 days. The circle only becomes visible for the correlations that present a level of significance ≤ 0.01. The blue color indicates a positive and the red color a negative correlation. BMI: body mass index; DFLac, difucosyllactose; DFLNH, difucosyllacto-N-hexaose; DFLNT, difucosyllacto-N-tetrose; DSLNH, disialyllacto-N-hexaose; DSLNT, disialyllacto-N-tetraose; FDSLNH, fucodisialyllacto-N-hexaose; FLNH, fucosyllacto-N-hexaose; HMO, human milk oligosaccharide; LNFP, lacto-N-fucopentaose; LNH, lacto-N-hexaose; LNnT, lacto-N-neotetraose; LNT, lacto-N-tetrose; LSTb, sialyl-lacto-N-tetraose b; LSTc, sialyl-lacto-N-tetraose c; 2′FL, 2′-fucosyllactose; 3FL, 3-fucosyllactose; 3′SL, 3′-sialyllactose; 6′SL, 6′-sialyllactose.

**Figure 6 nutrients-12-00790-f006:**
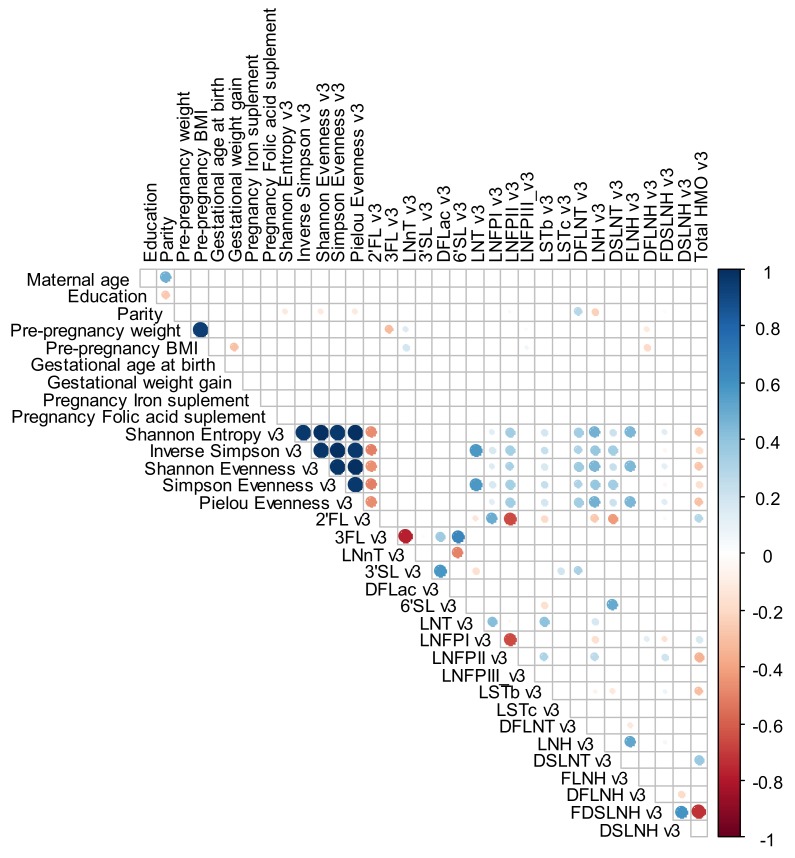
Spearman rank correlations between maternal demographic, anthropometric, reproductive and supplementary variables and individual HMO concentration in mmol/L and diversity and evenness at 88–119 days. The circle only becomes visible for the correlations that present a level of significance ≤ 0.01. The blue color indicates a positive and the red color a negative correlation. BMI: body mass index; DFLac, difucosyllactose; DFLNH, difucosyllacto-N-hexaose; DFLNT, difucosyllacto-N-tetrose; DSLNH, disialyllacto-N-hexaose; DSLNT, disialyllacto-N-tetraose; FDSLNH, fucodisialyllacto-N-hexaose; FLNH, fucosyllacto-N-hexaose; HMO, human milk oligosaccharide; LNFP, lacto-N-fucopentaose; LNH, lacto-N-hexaose; LNnT, lacto-N-neotetraose; LNT, lacto-N-tetrose; LSTb, sialyl-lacto-N-tetraose b; LSTc, sialyl-lacto-N-tetraose c; 2′FL, 2′-fucosyllactose; 3FL, 3-fucosyllactose; 3′SL, 3′-sialyllactose; 6′SL, 6′-sialyllactose.

**Table 1 nutrients-12-00790-t001:** Demographic and anthropometric profile and supplement use of women from the Brazilian cohort, Rio de Janeiro, Brazil.

Variables	Participants							
	With Samples (*n* = 101)				Without Samples (*n* = 46)			
	Median	IQR	Mean	SD	Median	IQR	Mean	SD
Maternal age (years) ^1^	26.1	22.8–31.6	27.2	5.5	27.8	22.8–30.4	26.9	5.4
Education (years) ^1^	12.0	10.0–12.0	10.9	2.8	10.5	8.2–12.0	10.5	2.6
Parity (livebirths) ^1^	0.0	0.0–1.0	0.7	0.8	0.5	0.0–1.0	0.8	1.1
Pre-pregnancy weight (kg) ^2^	63.0	55.0–72.0	65.1	14.8	61.5 ^5^	57.2–70.1 ^5^	64.2 ^5^	14.0 ^5^
Gestational age at birth (weeks)	39.4	38.9–40.6	39.5	1.4	39.7 ^6^	39.1–40.4 ^6^	39.5 ^6^	1.5 ^6^
Gestational weight gain (kg)	12.4	8.6–15.5	12.4	6.2	11.7^7^	8.7–18.0 ^7^	12.8 ^7^	7.9 ^7^
Pre-pregnancy body mass index (BMI) (kg/m^2^) ^3^	24.4 ^4^	21.3–28.8 ^4^	25.7 ^4^	5.7 ^4^	22.9 ^8^	21.8–29.6 ^8^	24.7 ^8^	4.9 ^8^
	***n* (%)**							
Pre-pregnancy BMI categories (kg/m^2^)								
Underweight (<18.5)	3 (3.2) ^4^				1 (6.7) ^8^			
Normal weight (18.5–24.9)	48 (50.5) ^4^				8 (53.3) ^8^			
Overweight (≥25.0)	44 (46.3) ^4^				6 (40.0) ^8^			
Iron supplement use (yes) ^9^	39 (42.9)				15 (35.7)			
Folic acid supplement (yes) ^9^	79 (87.8)				40 (95.2)			

IQR: interquartile range. Data of the participants with and without samples were not different according to Mann–Whitney, *p* < 0.05. ^1^ Baseline information. ^2^ Pre-pregnancy weight was self-reported. ^3^ Pre-pregnancy body mass index (BMI) was calculated with measured height at 28–50 days postpartum and pre-pregnancy weight. ^4^ Due to missing data, *n* = 95. ^5^ Due to missing data, *n* = 44. ^6^ Due to missing data, *n* = 35. ^7^ Due to missing data, *n* = 41. ^8^ Due to missing data, *n* = 15. ^9^ During pregnancy.

**Table 2 nutrients-12-00790-t002:** Time variation of human milk oligosaccharides (HMOs) concentrations in all women in the Brazilian cohort. Rio de Janeiro, Brazil (*n* = 101).

HMOs	All Women—Median (IQR) (mmol/L) Days Postpartum (*n*)		
	2–8 (*n* = 52)	28–50 (*n* = 75)	88–119 (*n* = 46)
**Fucosylated or Sialylated Lactose**			
2′FL	5.04 (4.02–6.48)	5.04 (3.31–5.85)	4.26 (2.96–6.16)
3FL	0.27 ^b^ (0.19–0.45)	0.36 ^c^ (0.25–0.84)	1.88 ^bc^ (0.39–5.29)
DFLac	0.25 ^ab^ (0.17–0.35)	0.37 ^ac^ (0.27–0.49)	0.44 ^bc^ (0.37–0.57)
3′SL	0.32 ^b^ (0.23–0.43)	0.38 ^c^ (0.28–0.60)	0.54 ^bc^ (0.43–0.70)
6′SL	0.37 ^ab^ (0.22–0.58)	0.63 ^ac^ (0.48–0.94)	0.49 ^bc^ (0.37–0.74)
**Non-Fucosylated, Non-Sialylated HMOs**			
LNT	1.40 (1.08–2.27)	1.45 (1.04–2.00)	1.36 (1.03–1.71)
LNnT	0.58 ^b^ (0.30–1.18)	0.28 ^c^ (0.18–0.42)	0.33 ^bc^ (0.17–0.58)
LNH	0.06 ^ab^ (0.03–0.11)	0.08 ^ac^ (0.06–0.14)	0.07 ^bc^ (0.04–0.09)
**Fucosylated, Non-Sialylated HMOs**			
LNFP I	2.66 ^b^ (2.04–3.48)	1.56 ^c^ (0.85–2.36)	0.76 ^bc^ (0.47–1.58)
LNFP II	0.77 ^ab^ (0.60–1.08)	1.12 ^ac^ (0.90–1.58)	1.69 ^bc^ (1.42–2.26)
LNFP III	0.08 ^b^ (0.07–0.12)	0.07 ^c^ (0.05–0.10)	0.04 ^bc^ (0.02–0.06)
DFLNT	1.47 ^b^ (0.62–1.81)	1.42 ^c^ (0.87–1.80)	1.20 ^bc^ (0.93–1.57)
FLNH	0.11 ^ab^ (0.05–0.16)	0.13 ^ac^ (0.08–0.22)	0.10 ^bc^ (0.07–0.14)
DFLNH	0.13 (0.05–0.21)	0.16 (0.09–0.25)	0.12 (0.04–0.21)
**Non-Fucosylated, Sialylated HMOs**			
LSTb	0.09 (0.07–0.12)	0.11 (0.07–0.15)	0.07 (0.05–0.11)
LSTc	0.77 ^b^ (0.63–0.96)	0.27 ^c^ (0.18–0.35)	0.10 ^bc^ (0.07–0.15)
DSLNT	0.58 ^b^ (0.38–0.74)	0.30 ^c^ (0.10–0.41)	0.18 ^bc^ (0.03–0.30)
DSLNH	0.22 ^ab^ (0.11–0.27)	0.22 ^ac^ (0.15–0.33)	0.09 ^bc^ (0.05–0.13)
**Fucosylated, Sialylated HMOs**			
FDSLNH	0.06 ^ab^ (0.04–0.12)	0.17 ^ac^ (0.09–0.31)	0.16 ^bc^ (0.09–0.24)
Total	16.6 ^b^ (15.66–17.36)	15.48 ^c^ (14.73–16.42)	16.79 ^bc^ (15.09–17.43)

IQR: interquartile range. ^a^ indicates that the HMO concentration at visit 1 is significantly different (*p* < 0.05) from the concentration at visit 2, ^b^ indicates that the HMO concentration at visit 1 is significantly different (*p* < 0.05) from the concentration at visit 3 and ^c^ indicates that the HMO concentration at visit 2 is significantly different (*p* < 0.05) from the concentration at visit 3 according to Friedman post-hoc test. DFLac, difucosyllactose; DFLNH, difucosyllacto-N-hexaose; DFLNT, difucosyllacto-N-tetrose; DSLNH, disialyllacto-N-hexaose; DSLNT, disialyllacto-Ntetraose; FDSLNH, fucodisialyllacto-N-hexaose; FLNH, fucosyllacto-N-hexaose; HMO, human milk oligosaccharide; LNFP, lacto-N-fucopentaose; LNH, lacto-N-hexaose; LNnT, lacto-N-neotetraose; LNT, lacto-N-tetrose; LSTb, sialyl-lacto-N-tetraose b; LSTc, sialyl-lacto-N-tetraose c; 2′FL, 2′-fucosyllactose; 3FL, 3-fucosyllactose; 3′SL, 3′-sialyllactose; 6′SL, 6′-sialyllactose.

**Table 3 nutrients-12-00790-t003:** Time variations in HMO group concentrations in women from the Brazilian cohort. Rio de Janeiro, Brazil (*n* = 101).

Variable (mmol/L)	All Women			Secretors Women			Non-Secretors Women		
	Days Postpartum (*n*)								
	2–8 (*n* = 52)	28–50 (*n* = 75)	88–119 (*n* = 46)	2–8 (*n* = 46)	28–50 (*n* = 68)	88–119 (*n* = 42)	2–8 (*n* = 6)	28–50 (*n* = 7)	88–119 (*n* = 4)
	Median (IQR)								
HMO-Bound Sialic Acid (All sialic acid moieties bound to HMOs)	3.45 ^a^ (3.07–3.90)	3.14 ^ac^ (2.64–3.64)	2.40 ^c^ (1.97–2.88)	3.45 ^x^ (3.08–3.88)	3.00 ^xz^ (2.64–3.55)	2.39 ^z^ (1.92–2.88)	3.33 (3.08–3.85)	3.85 (3.04–4.14)	2.61 (2.34–2.89)
HMO-Bound Fucose (All fucose moieties bound to HMOs)	13.59 ^b^ (12.63–14.67)	13.436 ^c^ (11.86–14.48)	15.04 ^bc^ (12.91–16.00)	13.68 ^y^ (12.84–14.97)	13.63 ^z^ (12.42–14.58)	15.29 ^yz^ (13.22–16.09)	3.77 (3.26–4.25)	5.26 (3.82–5.60)	8.09 (7.39–9.12)
Small HMOs (2′FL + 3FL + 3′SL + 6′SL)	6.26 ^b^ (4.77–7.90)	6.97 ^c^ (5.79–7.96)	8.58 ^bc^ (7.38–10.54)	6.66 (5.38–8.09)	7.11 (6.22–8.11)	8.86 (7.63–10.87)	0.50 (0.46–0.92)	1.53 (0.83–1.74)	1.84 (1.42–3.58)
Type 1 (LNT + LNFP I + LNFP II + LSTb + DSLNT)	5.80 ^a^ (5.30–6.58)	4.95 ^ac^ (4.14–5.93)	4.60 ^c^ (3.91–5.44)	5.73 ^x^ (5.30–6.48)	4.92 ^xz^ (4.15–5.94)	4.50 ^z^ (3.78–5.48)	6.26 (5.85–6.97)	5.14 (4.46–5.69)	4.73 (4.43–4.98)
Type 2 (LNnT + LNFP III + LSTc)	1.51 ^a^ (1.19–1.94)	0.65 ^ac^ (0.48–0.84)	0.50 ^c^ (0.29–0.70)	1.51 ^x^ (1.20–1.96)	0.65 ^xz^ (0.49–0.83)	0.55 ^z^ (0.31–0.76)	1.40 (1.19–1.65)	0.65 (0.43–0.93)	0.34 (0.24–0.45)
Alpha 1,2 (LNFP I + 2′FL)	7.94 (5.87–9.80)	6.67 (4.81–8.23)	5.36 (3.80–7.21)	8.66 ^x^ (7.23–10.11)	6.98 ^xz^ (5.59–8.37)	5.53 ^z^ (4.22–7.84)	0.12 (0.09–0.13)	0.14 (0.12–0.18)	0.17 (0.14–0.18)
Alpha 1,3 (LNFP III + 3FL)	0.37 ^b^ (0.28–0.55)	0.44 ^c^ (0.32–0.92)	1.90 ^bc^ (0.42–5.30)	0.38 ^y^ (0.30–0.55)	0.46 ^z^ (0.34–1.02)	2.27 ^yz^ (0.42–5.30)	0.22 (0.14–0.26)	0.24 (0.20–0.37)	1.07 (0.72–2.70)
Alpha 2,6 (LSTc + 6′SL)	1.28 ^a^ (0.97–1.68)	0.92 ^ac^ (0.78–1.27)	0.60 ^c^ (0.50–0.87)	1.29 ^x^ (1.00–1.68)	0.935 ^x^ (0.79–1.27)	0.61 (0.53–0.87)	0.92 (0.65–1.50)	0.82 (0.53–1.15)	0.46 (0.34–0.65)

IQR: interquartile range. ^a^ indicates that the HMO concentration at visit 1 is significantly different (*p* < 0.05) from the concentration at visit 2, ^b^ indicates that the HMO concentration at visit 1 is significantly different (*p* < 0.05) from the concentration at visit 3, ^c^ indicates that the HMO concentration at visit 2 is significantly different (*p* < 0.05) from the concentration at visit 3, ^x^ indicates that the HMO concentration at visit 1 is significantly different (*p* < 0.05) from the concentration at visit 2, ^y^ indicates that the HMO concentration at visit 1 is significantly different (*p* < 0.05) from the concentration at visit 3 and ^z^ indicates that the HMO concentration at visit 2 is significantly different (*p* < 0.05) from the concentration at visit 3 according to Friedman post-hoc test. DFLac, difucosyllactose; DFLNH, difucosyllacto-N-hexaose; DFLNT, difucosyllacto-N-tetrose; DSLNH, disialyllacto-N-hexaose; DSLNT, disialyllacto-N-tetraose; FDSLNH, fucodisialyllacto-N-hexaose; FLNH, fucosyllacto-N-hexaose; HMO, human milk oligosaccharide; LNFP, lacto-N-fucopentaose; LNH, lacto-N-hexaose; LNnT, lacto-N-neotetraose; LNT, lacto-N-tetrose; LSTb, sialyl-lacto-N-tetraose b; LSTc, sialyl-lacto-N-tetraose c; 2′FL, 2′-fucosyllactose; 3FL, 3-fucosyllactose; 3′SL, 3′-sialyllactose; 6′SL, 6′-sialyllactose.

**Table 4 nutrients-12-00790-t004:** Variation in HMO diversity and evenness indexes among women from Brazilian cohort. Rio de Janeiro, Brazil.

Indexes	Follow Up Visits in Days Postpartum		
	2–8 (*n* = 52)	28–50 (*n* = 75)	88–119 (*n* = 46)
Shannon entropy	2.152 ^ab^ (2.042–2.239)	2.236 ^ac^ (2.111–2.295)	2.073 ^bc^ (1.988– 2.189)
Inverse Simpson	5.764 (4.778–6.671)	5.890 (4.992–6.934)	5.310 (4.525–6.238)
Shannon evenness	0.730 ^ab^ (0.697–0.760)	0.760 ^ac^ (0.715–0.780)	0.700 ^bc^ (0.673–0.740)
Simpson evenness	0.303 (0.251–0.351)	0.310 (0.263–0.365)	0.279 (0.238–0.328)
Pielou evenness	0.731 ^ab^ (0.694–0.760)	0.759 ^ac^ (0.717–0.780)	0.704 ^bc^ (0.675–0.744)

All values are median and interquartile range. HMO: human milk oligosaccharide. ^a^ indicates that the HMO concentration at visit 1 is significantly different (*p* < 0.05) from the concentration at visit 2, ^b^ indicates that the HMO concentration at visit 1 is significantly different (*p* < 0.05) from the concentration at visit 3 and ^c^ indicates that the HMO concentration at visit 2 is significantly different (*p* < 0.05) from the concentration at visit 3 according to Friedman post-hoc test.

**Table 5 nutrients-12-00790-t005:** NMF scores according to time—postpartum period and general (for all times) for individual HMOs.

	Days Postpartum			
HMOs	2–8 (*n* = 52)	28–50 (*n* = 75)	88–119 (*n* = 46)	All Women (*n* = 101)
**Fucosylated or Sialylated Lactose**				
2′FL	0.38	0.37	1.00	0.80
3FL	0.36	0.74	0.89	0.01
DFLac	0.03	0.06	0.10	0.02
3′SL	0.05	0.15	0.03	0.01
6′SL	0.42	0.12	0.10	0.29
**Non-Fucosylated, Non-Sialylated HMOs**				
LNT	0.48	0.27	0.03	0.14
LNnT	0.37	0.04	0.41	0.89
LNH	0.38	0.11	0.13	0.28
**Fucosylated, Non-Sialylated HMOs**				
LNFP I	0.41	0.62	0.76	0.43
LNFP II	0.53	0.30	0.16	0.14
LNFP III	0.03	0.04	0.09	0.70
DFLNT	0.44	0.51	0.45	0.46
FLNH	0.10	0.05	0.01	0.43
DFLNH	0.27	0.38	0.38	0.35
**Fucosylated, Non-Sialylated HMOs**				
LSTb	0.45	0.07	0.13	0.12
LSTc	0.08	0.01	0.05	0.84
DSLNT	0.37	0.03	0.48	0.39
DSLNH	0.38	0.04	0.19	0.37
**Fucosylated, Sialylated HMOs**				
FDSLNH	0.48	0.31	0.57	0.38

NMF scores represent the probability of contribution to (and importance of) a specified HMO variable to the basis component. HMO: human milk oligosaccharide. 2′FL: 2′-fucosyllactose. 3FL: 3-fucosyllactose. LNnT: lacto-N-neotetraose. 3′SL: 3′-sialyllactose. DFLac: difucosyllactose. 6′SL: 6′-sialyllactose. LNT: lacto-N-tetrose. LNFP: lacto-N-fucopentaose. LSTb: sialyl-lacto-N-tetraose b. LSTc: sialyl-lacto-N-tetraose c. DFLNT: difucosyllacto-N-tetrose. LNH: lacto-N-hexaose. DSLNT: disialyllacto-Ntetraose. FLNH: fucosyllacto-N-hexaose. DFLNH: difucosyllacto-N-hexaose. FDSLNH: fucodisialyllacto-N-hexaose. DSLNH: disialyllacto-N-hexaose.

**Table 6 nutrients-12-00790-t006:** NMF scores according to maternal age quartiles for individual HMOs.

	Maternal Age in Years ^1^			
HMOs	18.0–22.5 (*n* = 26)	22.7–26.6 (*n* = 27)	26.9–31.4 (*n* = 21)	31.5–40.0 (*n* = 27)
**Fucosylated or Sialylated Lactose**				
2′FL	0.54	0.60	0.99	0.44
3FL	0.16	0.01	0.06	0.20
DFLac	0.10	0.01	0.01	0.21
3′SL	0.12	0.21	0.01	0.17
6′SL	0.18	0.46	0.20	0.14
**Non-Fucosylated, Non-Sialylated HMOs**				
LNT	0.03	0.63	0.54	0.54
LNnT	0.65	0.79	0.92	0.64
LNH	0.22	0.09	0.39	0.03
**Fucosylated, Non-Sialylated HMOs**				
LNFP I	0.04	0.28	0.27	0.03
LNFP II	0.54	0.14	0.09	0.37
LNFP III	0.46	0.24	0.40	0.40
DFLNT	0.55	0.12	0.31	0.28
FLNH	0.47	0.30	0.43	0.08
DFLNH	0.37	0.02	0.29	0.44
**Non-Fucosylated, Sialylated HMOs**				
LSTb	0.02	0.06	0.40	0.05
LSTc	0.52	0.74	0.83	0.53
DSLNT	0.10	0.20	0.45	0.19
DSLNH	0.55	0.47	0.29	0.19
**Fucosylated, Sialylated HMOs**				
FDSLNH	0.24	0.12	0.25	0.21

NMF scores represent the probability of contribution to (and importance of) a specified HMO variable to the basis component. ^1^ Baseline information. HMO: human milk oligosaccharide. 2′FL: 2′-fucosyllactose. 3FL: 3-fucosyllactose. LNnT: lacto-N-neotetraose. 3′SL: 3′-sialyllactose. DFLac: difucosyllactose. 6′SL: 6′-sialyllactose. LNT: lacto-N-tetrose. LNFP: lacto-N-fucopentaose. LSTb: sialyl-lacto-N-tetraose b. LSTc: sialyl-lacto-N-tetraose c. DFLNT: difucosyllacto-N-tetrose. LNH: lacto-N-hexaose. DSLNT: disialyllacto-Ntetraose. FLNH: fucosyllacto-N-hexaose. DFLNH: difucosyllacto-N-hexaose. FDSLNH: fucodisialyllacto-N-hexaose. DSLNH: disialyllacto-N-hexaose.

**Table 7 nutrients-12-00790-t007:** NMF scores according to pre-pregnancy BMI for individual HMOs.

	Pre-Pregnancy BMI in kg/m^2^			
HMOs	Underweight (<18.5; *n* = 3)	Normal Weight (18.5–24.9; *n* = 48)	Overweight (25.0–29.9; *n* = 29)	Obesity (>29.9; *n* = 15)
**Fucosylated or Sialylated Lactose**				
2′FL	0.16	0.61	0.99	1.00
3FL	0.10	0.05	0.16	0.04
DFLac	0.10	0.00	0.01	0.22
3′SL	0.57	0.02	0.01	0.01
6′SL	0.03	0.35	0.16	0.57
**Non-Fucosylated, Non-Sialylated HMOs**				
LNT	0.10	0.32	0.62	0.11
LNnT	0.10	0.80	0.84	0.90
LNH	0.04	0.31	0.19	0.23
**Fucosylated, Non-Sialylated HMOs**				
LNFP I	0.11	0.38	0.56	0.32
LNFP II	0.18	0.11	0.12	0.06
LNFP III	0.30	0.71	0.55	0.47
DFLNT	0.15	0.48	0.23	0.22
FLNH	0.31	0.28	0.35	0.49
DFLNH	0.28	0.22	0.10	0.31
**Non-Fucosylated, Sialylated HMOs**				
LSTb	0.07	0.06	0.33	0.08
LSTc	0.18	0.86	0.85	0.76
DSLNT	0.21	0.58	0.21	0.16
DSLNH	0.03	0.37	0.38	0.39
**Fucosylated, Sialylated HMOs**				
FDSLNH	0.06	0.30	0.21	0.31

NMF scores represent the probability of contribution to (and importance of) a specified HMO variable to the basis component. HMO: human milk oligosaccharide. 2′FL: 2′-fucosyllactose. 3FL: 3-fucosyllactose. LNnT: lacto-N-neotetraose. 3′SL: 3′-sialyllactose. DFLac: difucosyllactose. 6′SL: 6′-sialyllactose. LNT: lacto-N-tetrose. LNFP: lacto-N-fucopentaose. LSTb: sialyl-lacto-N-tetraose b. LSTc: sialyl-lacto-N-tetraose c. DFLNT: difucosyllacto-N-tetrose. LNH: lacto-N-hexaose. DSLNT: disialyllacto-Ntetraose. FLNH: fucosyllacto-N-hexaose. DFLNH: difucosyllacto-N-hexaose. FDSLNH: fucodisialyllacto-N-hexaose. DSLNH: disialyllacto-N-hexaose.

**Table 8 nutrients-12-00790-t008:** NMF scores according to parity for individual HMOs.

	Parity in Number of Children ^1^			
HMOs	0 (*n* = 53)	1 (*n* = 31)	2 (*n* = 13)	≥3 (*n* = 3)
**Fucosylated or Sialylated Lactose**				
2′FL	0.98	0.99	0.60	0.27
3FL	0.02	0.01	0.11	0.44
DFLac	0.03	0.05	0.13	0.45
3′SL	0.06	0.01	0.02	0.41
6′SL	0.30	0.34	0.37	0.08
**Non-Fucosylated, Non-Sialylated HMOs**				
LNT	0.10	0.54	0.14	0.06
LNnT	0.90	0.96	0.82	0.50
LNH	0.26	0.36	0.17	0.38
**Fucosylated, Non-Sialylated HMOs**				
LNFP I	0.47	0.40	0.34	0.16
LNFP II	0.10	0.04	0.21	0.01
LNFP III	0.80	0.17	0.79	0.10
DFLNT	0.48	0.19	0.21	0.03
FLNH	0.46	0.39	0.19	0.39
DFLNH	0.35	0.14	0.18	0.19
**Non-Fucosylated, Sialylated HMOs**				
LSTb	0.03	0.13	0.11	0.45
LSTc	0.83	0.88	0.73	0.46
DSLNT	0.34	0.20	0.46	0.29
DSLNH	0.37	0.44	0.33	0.48
**Fucosylated, Sialylated HMOs**				
FDSLNH	0.40	0.04	0.35	0.23

NMF scores represent the probability of contribution to (and importance of) a specified HMO variable to the basis component. ^1^ Baseline information. HMO: human milk oligosaccharide. 2′FL: 2′-fucosyllactose. 3FL: 3-fucosyllactose. LNnT: lacto-N-neotetraose. 3′SL: 3′-sialyllactose. DFLac: difucosyllactose. 6′SL: 6′-sialyllactose. LNT: lacto-N-tetrose. LNFP: lacto-N-fucopentaose. LSTb: sialyl-lacto-N-tetraose b. LSTc: sialyl-lacto-N-tetraose c. DFLNT: difucosyllacto-N-tetrose. LNH: lacto-N-hexaose. DSLNT: disialyllacto-Ntetraose. FLNH: fucosyllacto-N-hexaose. DFLNH: difucosyllacto-N-hexaose. FDSLNH: fucodisialyllacto-N-hexaose. DSLNH: disialyllacto-N-hexaose.

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
