# Peer review of "Human Milk Oligosaccharide Profile Variation Throughout Postpartum in Healthy Women in a Brazilian Cohort"

_nutrients, 2020, doi:10.3390/nu12030790_

Round 1

Reviewer 1 Report

No comments

Author Response

Rio de Janeiro, Brazil
March 6th, 2020
Reference number nutrients-733733
Dear reviewer,     
I would like to acknowledge the receipt of comments regarding the manuscript entitled ‘Human milk oligosaccharide profile variation throughout postpartum in healthy women in a Brazilian cohort’. The authors have revised the manuscript according the reviewers’ comments and responded to the questions raised.
 The main change performed is listed below:
  1. The results comprising the 15 mothers who provided multiple samples across the three timepoints were analyzed as requested by reviewer 2 and 3. The findings based on this subset reaffirm the conclusions of the original submission, as the vast majority of the results remained the same when compared with the overall dataset. We have included a paragraph in the results and one in the discussion section addressing these comparisons (sensitivity analyses). However, we opted to present the results for the complete sample size.
We are available for any future concerns you may have.          

Sincerely,
Professor Gilberto Kac
Universidade Federal do Rio de Janeiro
Avenida Carlos Chagas Filho 367
CCS – Bloco J – 2º andar, sala 29
Cidade Universitária – Ilha do Fundão
Rio de Janeiro, RJ, 21941-590, Brasil.
Telephone: (55-21) 39386595/ (55-21) 39380253
E-mail: [email protected]
Response to Reviewer 1 Comments
         We thank very much the reviewer’s and the editor of the Nutrients for their kind and useful comments and suggestions that definitely helped to improve the quality of our manuscript. In the responses below, author comments are in red and bold.
Thank you for taking the time to review the submitted manuscript.

Reviewer 2 Report

This is an interesting study by Ferreira and colleagues that chronologically examined the variations in HMO compositions in post-partum women within a Brazilian cohort. Despite the stated limitations in sample size compared to previous studies that looked at similar outcomes, several intriguing points are quite remarkable. Within the 3 months post-partum period, the group showed that 1) while concentrations of HMOs decrease, the molar concentrations of HMOs elevate, 2) HMOs diversity shift over course of lactation and is particularly high at 28-50 days, 3) HMOs were correlated with BMI and pre-pregnancy weight but with some differences in directionality of correlations. The findings do provide interesting interpretation about the host factors that mediate HMOs composition as well as physiologic variations in HMOs post-partum. Some points to consider:

1) Within the 15 mothers who provided multiple samples across 3 timepoints, have the group attempted to recapitulate some of the findings to see if the same trend exist for those mothers? In a smaller sample cohort, this may help control for other cofounders that may not have been accounted for in the baseline characteristics, particularly when we have no information about the general health, medications, anthropomorphic changes post-partum and etc. 

2) While there were differences in weights/BMI, do we have any information on the diets of the mothers pre, peri and post-partum? 

Author Response

Rio de Janeiro, Brazil
March 6th, 2020
Reference number nutrients-733733
Dear reviewer,     

I would like to acknowledge the receipt of comments regarding the manuscript entitled ‘Human milk oligosaccharide profile variation throughout postpartum in healthy women in a Brazilian cohort’. The authors have revised the manuscript according the reviewers’ comments and responded to the questions raised.
 The main change performed is listed below:

  1. The results comprising the 15 mothers who provided multiple samples across the three timepoints were analyzed as requested by reviewer 2 and 3. The findings based on this subset reaffirm the conclusions of the original submission, as the vast majority of the results remained the same when compared with the overall dataset. We have included a paragraph in the results and one in the discussion section addressing these comparisons (sensitivity analyses). However, we opted to present the results for the complete sample size.

We are available for any future concerns you may have.          

Sincerely,
Professor Gilberto Kac
Universidade Federal do Rio de Janeiro
Avenida Carlos Chagas Filho 367
CCS – Bloco J – 2º andar, sala 29
Cidade Universitária – Ilha do Fundão
Rio de Janeiro, RJ, 21941-590, Brasil.
Telephone: (55-21) 39386595/ (55-21) 39380253
E-mail: [email protected]
Response to Reviewer 2 Comments
         We thank very much the reviewer’s and the editor of the Nutrients for their kind and useful comments and suggestions that definitely helped to improve the quality of our manuscript. In the responses below, author comments are in red and bold.
1. Within the 15 mothers who provided multiple samples across 3 timepoints, have the group attempted to recapitulate some of the findings to see if the same trend exist for those mothers? In a smaller sample cohort, this may help control for other cofounders that may not have been accounted for in the baseline characteristics, particularly when we have no information about the general health, medications, anthropomorphic changes post-partum and etc.
We reproduced the complete analyses using only a subset of 15 women with samples in all time periods. The majority of the results confirm the original findings and reaffirm the conclusions of the study, such as smaller total HMO concentration at last visit in g/L and obese women with 2'fucosyllactose as the main contributor to the HMO profile. However, it is important to highlight that some minor differences were found, and this can be probably be attributable to the small sample size of the subset analyzed. The most abundant HMO on non-secretor woman in this subset at the last visit was different from the original one (3’-sialyllactose). Spearman rank significant correlations were observed between maternal age and FLNH at visit 2 and 3. Finally, maternal characteristics, such as primiparity and normal weight and excess weight, showed a different HMO as the main contributor in the subgroup, when compared with the original study results.
 2. While there were differences in weights/BMI, do we have any information on the diets of the mothers pre, peri and post-partum?
The main objective of this manuscript was to describe the profile of HMOs and the role of selected maternal characteristics on this profile. Therefore, despite the presence of dietary data on our study, it is our understanding that the role of diet on HMO deserves to be explored on a sole manuscript. 

Reviewer 3 Report

In this work, Ferreira et al. analyzed a group of 19 human milk oligosaccharides in breast milk samples collected at three different lactation time points from 101 Brazilian women. This was claimed to be the first study of HMO concentrations at different time points during lactation in Latin American women. The total weight concentration of HMOs in this cohort decreased over the course of lactation as expected, while interestingly, the total molar concentration slightly increased. Secretor status determined by the abundances of 2’FL and LNFP I showed that 89.1% of the women were secretors. The differences between secretors and nonsecretors and their respective changes throughout postpartum were compared within this cohort and against the literature. The correlations between maternal sociodemographic and anthropometric characteristics of these healthy women and their HMO composition were also investigated. Overall, the study was well designed, and the manuscript is nicely written. However, there are a few points that need to be clarified/revised to strengthen the manuscript.

  1. Ref 38 was used to describe the method for HMO analysis. However, in this reference, there is not a complete description of the method either. The authors either need to add more relevant references or include in the current manuscript more details about the 2-AB derivatization, SPE cleanup, and HPLC separation methods.
  2. Table 2 is titled “Time variation of HMOs concentrations in total and according to Secretor status of women followed on Brazilian cohort. Rio de Janeiro, Brazil (n=101).”, but no comparison based on secretor status is shown in this Table. The authors also indicated there is a table in the Supplementary Materials with the same title. Was this a mistake?
  3. Figure 2 caption is confusing because (a)/(d), (b)/(e) and (c)/(f) show all women, secretors and nonsecretors, respectively, not 2 – 8 days, 28 – 50 days, and 88 – 119 days. There is also a typo. It should be 2 – 8 (a)/(d) instead of 28.
  4. There were 15 subjects that had samples for all three time points. Were these samples treated as independent inputs for statistical analysis? Did the authors analyze these longitudinal samples separately to study the HMO changes over lactation?
  5. Ref 34, the title in the citation is not complete.

Author Response

Rio de Janeiro, Brazil
March 6th, 2020
Reference number nutrients-733733

Dear reviewer,     
I would like to acknowledge the receipt of comments regarding the manuscript entitled ‘Human milk oligosaccharide profile variation throughout postpartum in healthy women in a Brazilian cohort’. The authors have revised the manuscript according the reviewers’ comments and responded to the questions raised.
 The main change performed is listed below:
  1. The results comprising the 15 mothers who provided multiple samples across the three timepoints were analyzed as requested by reviewer 2 and 3. The findings based on this subset reaffirm the conclusions of the original submission, as the vast majority of the results remained the same when compared with the overall dataset. We have included a paragraph in the results and one in the discussion section addressing these comparisons (sensitivity analyses). However, we opted to present the results for the complete sample size.
We are available for any future concerns you may have.          

Sincerely,
Professor Gilberto Kac
Universidade Federal do Rio de Janeiro
Avenida Carlos Chagas Filho 367
CCS – Bloco J – 2º andar, sala 29
Cidade Universitária – Ilha do Fundão
Rio de Janeiro, RJ, 21941-590, Brasil.
Telephone: (55-21) 39386595/ (55-21) 39380253
E-mail: [email protected]
Response to Reviewer 3 Comments
         We thank very much the reviewer’s and the editor of the Nutrients for their kind and useful comments and suggestions that definitely helped to improve the quality of our manuscript. In the responses below, author comments are in red and bold.
1. Ref 38 was used to describe the method for HMO analysis. However, in this reference, there is not a complete description of the method either. The authors either need to add more relevant references or include in the current manuscript more details about the 2-AB derivatization, SPE cleanup, and HPLC separation methods.
We provided additional details to describe the HMO analytical method and have also switched reference 38 for a reference with more detailed description of the HMO analytical procedure. The new reference can be found below:
Jantscher-Krenn, E.; Lauwaet, T.; Bliss, L.A.; Reed, S.L.; Gillin, F.D.; Bode, L. Human milk oligosaccharides reduce Entamoeba histolytica attachment and cytotoxicity in vitro. Br J Nutr 2012, 108, 1839-1846, doi:10.1017/S0007114511007392.
2. Table 2 is titled “Time variation of HMOs concentrations in total and according to Secretor status of women followed on Brazilian cohort. Rio de Janeiro, Brazil (n=101).”, but no comparison based on secretor status is shown in this Table. The authors also indicated there is a table in the Supplementary Materials with the same title. Was this a mistake? 
We apologize for the mistake. Both titles have been corrected and are shown below:
Table 2. Time variation of HMOs concentrations in all women followed on Brazilian cohort. Rio de Janeiro, Brazil (n=101).
Table S1. Time variation of HMOs concentrations according to Secretor status of women followed on Brazilian cohort. Rio de Janeiro, Brazil (n=101).
3. Figure 2 caption is confusing because (a)/(d), (b)/(e) and (c)/(f) show all women, secretors and nonsecretors, respectively, not 2 – 8 days, 28 – 50 days, and 88 – 119 days. There is also a typo. It should be 2 – 8 (a)/(d) instead of 28.
The figure 2 caption was corrected and presents (a)/(d), (b)/(e) and (c)/(f) for all women, secretors and non-secretors, respectively, as shown below:
Figure 2. HMO Absolute and relative abundance of women followed on Brazilian cohort. Rio de Janeiro, Brazil. Median of absolute total HMO (mmol/L) (a,b and c) at 2-8, 28-50 and 88-119 days and relative abundance (d, e and f) at 2-8, 28-50 and 88-119 days concentrations of all women, according Secretor status and follow up points.
4. There were 15 subjects that had samples for all three time points. Were these samples treated as independent inputs for statistical analysis? Did the authors analyze these longitudinal samples separately to study the HMO changes over lactation?
We reproduced the complete analyses using only a subset of 15 women with samples in all time periods. The majority of the results confirm the original findings and reaffirm the conclusions of the study, such as smaller total HMO concentration at last visit in g/L and obese women with 2'fucosylactose as the main contributor to the HMO profile. However, it is important to highlight that some minor differences were found, and this can be probably be attributable to the small sample size of the subset analyzed. The most abundant HMO on non-secretor woman in this subset at the last visit was different from the original one (3’-sialyllactose). Spearman rank significant correlations were observed between maternal age and FLNH at visit 2 and 3. Finally, maternal characteristics, such as primiparity and normal weight and excess weight, showed a different HMO as the main contributor in the subgroup, when compared with the original study results.
5. Ref 34, the title in the citation is not complete.
The reference was completed and can be found below:
Sprenger, N.; Lee, L.; De Castro, C.; Steenhout, P.; Thakkar, S. - Longitudinal change of selected human milk oligosaccharides and association to infants' growth, an observatory, single center, longitudinal cohort study. PLoS One 2017, 12, 0171814.
